

# The effect of cognitive training on the subjective perception of well-being in older adults

Vladimír Bureš[1], Pavel Čech[1], Jaroslava Mikulecká[1], Daniela Ponce[1] and Kamil Kuca[1,2]

[1] Faculty of Informatics and Management, University of Hradec Kralove, Hradec Kralove, Czech Republic
[2] Biomedical Research Centre, University Hospital Hradec Kralove, Hradec Kralove, Czech Republic

Corresponding author
Pavel Čech, pavel.cech@uhk.cz

## ABSTRACT

**Background.** There is a growing number of studies indicating the major consequences of the subjective perception of well-being on mental health and healthcare use. However, most of the cognitive training research focuses more on the preservation of cognitive function than on the implications of the state of well-being. This secondary analysis of data from a randomised controlled trial investigated the effects of individualised television-based cognitive training on self-rated well-being using the WHO-5 index while considering gender and education as influencing factors. The effects of cognitive training were compared with leisure activities that the elderly could be engaged in to pass time.

**Methods.** Cognitively healthy participants aged 60 years or above screened using the Mini-Mental State Examination (MMSE) and Major Depression Inventory (MDI) were randomly allocated to a cognitive training group or to an active control group in a single-blind controlled two-group design and underwent 24 training sessions. Data acquired from the WHO-5 questionnaire administered before and after intervention were statistically analysed using a mixed design model for repeated measures. The effect of individualised cognitive training was compared with leisure activities while the impact of gender and education was explored using estimated marginal means.

**Results.** A total of 81 participants aged $67.9 \pm 5.59$ [60–84] without cognitive impairments and absent of depression symptoms underwent the study. Participants with leisure time activities declared significantly higher scores compared to participants with cognitive training $M = 73.48 \pm 2.88$, 95% CI [67.74–79.22] vs $M = 64.13 \pm 3.034$, 95% CI [58.09–70.17] WHO-5 score. Gender and education were found to moderate the effect of cognitive training on well-being when compared to leisure activities. Females engaged in leisure activities in the control group reported higher by $M = 9.77 \pm 5.4$, 95% CI [−0.99–20.54] WHO-5 scores than females with the cognitive training regimen. Participants with high school education declared leisure activities to increase WHO-5 scores by $M = 14.59 \pm 5.39$, 95% CI [3.85–25.34] compared to individualised cognitive training.

**Discussion.** The findings revealed that individualised cognitive training was not directly associated with improvements in well-being. Changes in the control group indicated that involvement in leisure time activities, in which participants were partly free to choose from, represented more favourable stimulation to a self-perceived sense of well-being than individualised cognitive training. Results also supported the fact that gender and education moderated the effect of cognitive training on well-being. Females

and participants with high school education were found to be negatively impacted in well-being when performance connected with cognitive training was expected.

## INTRODUCTION

There is evidence that cognitive training has a positive effect on cognitive function, and the preservation of cognitive function is associated with a healthier lifestyle and greater well-being (*Henry et al., 2015*; *Llewellyn et al., 2008*; *WHO, 2011*). However, recent studies on computer-based cognitive training have focused mainly on improvements in cognitive function or transfer to other cognitive tasks (*Kueider et al., 2012*; *Maseda et al., 2013*; *Rizkalla, 2015*; *Shatil et al., 2014*; *Van Muijden, Band & Hommel, 2012*) rather than on the self-perceived emotional experience denoted as well-being. Therefore, the effect of cognitive training on psychological well-being remains unclear. Cognitive training is believed to positively influence well-being as a consequence of preserving the cognitive function (*Adamski et al., 2016*; *Innes et al., 2016*; *Smith, 2016*) but this relationship is assumed rather than supported by evidence. Correspondingly, not much is known about factors that might influence the observed effect of cognitive training on well-being.

The rationale for drawing attention to well-being is due to studies indicating the major consequences of well-being and of positive mental health for a healthier life, improved healthcare use and corresponding social outcomes (*Allerhand, Gale & Deary, 2014*; *Buiza et al., 2009*; *Walker & Lowenstein, 2009*). In health, well-being denotes optimal psychological functioning and experience (*Ryan & Deci, 2001*) and consists of the ability to be happy or at least contended and have the capacity that enhances self-esteem, maintains beliefs in personal efficacy, and promotes an optimistic view of the future (*Taylor & Brown, 1988*). Well-being is argued to complement functional and health status in assessing patients' quality of life and is considered as a proxy indicator of quality of life in response to a treatment, intervention and also to ageing (*Brown, Bowling & Flynn, 2004*).

Even normally aging older adults experience significant impairment in attentional tasks (*Getzmann, Golob & Wascher, 2016*; *Sperduti, Makowski & Piolino, 2016*), episodic memory (*Manenti et al., 2016*; *Tromp et al., 2015*) and working memory tasks (*Jost et al., 2011*; *Ko et al., 2014*), time perception (*Turgeon, Lustig & Meck, 2016*) and also speed of processing (*Ball et al., 2013*; *Elgamal, Roy & Sharratt, 2011*). If poorly prevented, these impairments might degrade the quality of life as well as the psychological well-being (*Pusswald et al., 2015*; *Stogmann et al., 2015*). Other studies have also indicated that cognitive decline in its early stages often goes unnoticed by individuals as the activities of daily living in familiar environments are not affected (*Calzà et al., 2015*; *Missotten et al., 2008*; *Ready et al., 2003*) and these studies further stress the importance of prevention in order to maintain cognitive abilities and well-being.

There is evidence that women feel worse than men in terms of subjective well-being (*Stevenson & Wolfers, 2009*), yet women are reported to declare higher levels of well-being when they experience similar conditions to men (*Senik, 2015*). The subjective well-being gap between women and men in high-income countries is explained by women having different uses of their time, lower expectations especially concerning their social role and richer social ties (*Graham & Chattopadhyay, 2013*; *Lun & Bond, 2016*). The gender gap in well-being could also be related to differences in mental health in men and women. According to the World Health Organization, gender is a critical determinant of mental health and mental illness (*WHO, 2015*). Unipolar depression, for instance, is twice as common in women and gender has also been found to be a significant predictor of psychotropic drug prescription (*Nana et al., 2015*; *Simoni-Wastila, 2000*).

Rarely is the impact of education on well-being in connection with cognitive training in elderly population addressed. Nevertheless, studies on cognitive training suggest that specific cognitive exercises are moderated by educational attainment. For instance, the study of *Willis & Caskie (2013)* reports that higher education is related to higher baseline scores in cognitive training exercises. However, participants with lower levels of education exhibited greater training effects in Letter and Word Series exercises (*Willis & Caskie, 2013*). Similarly, a recent study by *Clark et al. (2015)* shows that the effect of cognitive training on cognitive function does not differ in memory and reasoning exercises by educational attainment but it is significantly different in the speed of processing. The effect of speed of processing training is 50% greater in normal older adults with a less than complete high school education (*Clark et al., 2015*). In contrast, *Maseda et al. (2013)* point out that the effect of cognitive training was less pronounced for subjects with incomplete primary studies. Given the assumed link between improved cognitive function and improved well-being, the conjecture in the present study is that the education might moderate also the effect of cognitive training on well-being.

Recently, there were attempts to study well-being in relation to meditation (*Goyal et al., 2014*), yoga involvement (*Gaiswinkler & Unterrainer, 2016*), physical exercise such as running (*Skead & Rogers, 2016*) or whole body movements (*Ben-Soussan et al., 2015*), social engagement (*Rapacciuolo et al., 2016*) or intergeneration social support (*Tian, 2016*). These studies prove that well-being might be stimulated by consciously focusing on the positive side of life, by fostering an active and healthy life-style and by encouraging social ties. In respect to cognitive function *Allerhand, Gale & Deary (2014)* pointed out that higher levels of well-being are associated with better cognitive performance and also that above average cognitive function implies greater well-being. The positive bidirectional relationship of cognitive function and well-being supports the assumed effect of cognitive training. However, cognitive training studies do not prove this link. There are studies on cognitive training in which positive effect on well-being is often implicitly inferred based on improvements in cognitive function but not explicitly measured (*Adamski et al., 2016*; *Gates et al., 2011*; *Innes et al., 2016*; *Smith, 2016*). In a study by *Jansen & Dahmen-Zimmer (2012)* the effect of cognitive training on well-being is analysed in combination with motor and karate training. However, they used the Anxiety-Depression Scale instead of scales designed to measure well-being. It has already been pointed out that although depression

is associated with lower levels of well-being, there is increasing evidence that positive well-being and negative mental states are to some degree independent (*Allerhand, Gale & Deary, 2014*; *Huppert & Whittington, 2003*; *Isaacowitz & Smith, 2003*). There are also cognitive training studies that focus on people with already developing mental problems such as mild cognitive impairments (*Belleville et al., 2006*; *Bier et al., 2015*) or memory complaints (*Cohen-Mansfield et al., 2015*). A majority of the studies does not clearly explicate the effect of cognitive training on well-being in a population of healthy elderly adults. Therefore, the present research elaborates on the following research question: What will be the difference in the self-perceived sense of well-being of elderly people involved in individualised cognitive training in regard to gender or education? Studying the effect of cognitive training on the subjective perception of well-being might help in understanding how older people perceive the cognitive training as a part of their everyday lives and what the implications for training adherence are. The aim of this study is to investigate if there is a statically significant effect of individualised television-based cognitive training on self-perceived well-being based on the WHO-5 index while controlling for education or gender. The effect of cognitive training is compared with a control group engaged in leisure time activities that elderly adults can do to pass the time.

## MATERIALS AND METHODS

### Study design

The study design was based on secondary analysis of a randomised controlled trial conducted at the facilities of the University of Hradec Kralove, in the East Bohemia Region, Czech Republic. The trial was part of the project Vital Mind (European Union 7th Framework Programme, project number ICT-215387) that focused on migrating computer-based cognitive training to a digital television platform. The study was designed to consist of 24 sessions over the course of eight weeks, i.e., three sessions per week were administered to the participants. Each session was supposed to last 20–30 min, totalling 8–12 h. The sessions were held during the morning and afternoon (from 8 a.m. till 4 p.m.) in special rooms. There were four rooms operating in parallel in order to serve all the participants. Each session was assisted by trained personnel. The aim was to provide administrative and technical assistance. During the training part of the session, the assistant interfered only in situations in which the participants were in doubt about how to continue or were having problems with the technology. The European Commission granted ethical approval to carry out the study within the scope of the Vital Mind project by accepting the Ethical Road Map report (deliverable D8.6.1). The institutional ethical approval was granted by the chancellor's office of the University of Hradec Kralove (Vital Mind ref. no. 215387). The study was undertaken with the understanding and written informed consent of each participant and according to the World Medical Association Declaration of Helsinki ethical principles.

## Participants

The sample consisted of 140 older adults aged 60+ from the East Bohemia Region. The candidates were recruited via advertisements in all local newspapers and by a recruitment procedure at the University of Lifelong Learning. The University of Lifelong Learning is a public institute affiliated with the University of Hradec Kralove that provides further education to more than 1,500 attendees every year. During the recruitment procedure at the University of Lifelong Learning, information about the study was announced during regular courses and the contact details of prospective volunteers were collected. Volunteers were then invited for a brief introduction to the study after which they were asked to confirm their interest in participating in the study.

This study focused only on retired people who do not live in community centres or care homes. The eligibility assessment was conducted under the supervision of a psychologist using the Mini-Mental State Examination (MMSE) (*Folstein, Folstein & McHugh, 1975*) to test for cognitive impairment and Major Depression Inventory (MDI) (*Bech et al., 2001*; *Olsen et al., 2003*) to screen for depressive symptoms. The inclusion criteria were the following: age ≥ 60, MMSE ≥ median values adjusted for age and education reported by *Crum et al. (1993)*, absence of moderate to severe depressive symptoms based on ICD-10 algorithm (*Bech et al., 2001*), no assistance required in daily routines, no motor, visual and hearing impairments that would hinder the training. Based on the inclusion criteria, 81 participants were admitted to take part in the study. The participants were informed that finishing all planned sessions would be financially rewarded with €100 approx. to cover travel and other 'out-of-pocket' expenses.

Participants were randomly divided into two groups: an active control group (referred to as control group hereafter) and a cognitive group with leisure time activities and with individualised cognitive training respectively. The randomisation procedure was carried out using computer generated pseudorandom numbers assigned to each participant. The generated numbers were then randomly divided into the two given groups. To avoid placebo like effects and motivation distortion described by *Boot, Blakely & Simons (2011)* in which people in a cognitive group perform better because they are expected to do so, participants were not aware of the distinction between the groups and the aim of the research was generalised to both groups (single-blind controlled two-group design).

## Materials

The MMSE (*Folstein, Folstein & McHugh, 1975*) in the Czech version (*Brazdil, Ruta & Sobotka, 1995*) was used as a screening tool for cognitive impairment and the MDI (*Bech et al., 2001*; *Olsen et al., 2003*) was used to indicate depressive symptoms. Both the MMSE and MDI tests were administered by trained personnel (senior researchers) under the supervision of a psychologist. The MMSE cut-off score of 24 was suggested by *Folstein, Folstein & McHugh (1975)*, however, later studies showed that the MMSE score was moderated by age and education (*Crum et al., 1993*; *O'Bryant et al., 2008*). There have been no validation study with normative scores of MMSE stratified by age and years of education in Czech population, therefore, the present study adopted median values reported by *Crum et al. (1993)* as cut-off scores.

The MDI answers were dichotomised based on the ICD–10 algorithm in which moderate to severe depression is observed if at least two out of three core symptoms are present as well as at least four of the other seven items (*Bech et al., 2001*). The WHO-5 Well-Being Index was used for well-being screening. The WHO-5 Well-Being Index covering five positively worded items, related to positive mood, vitality and general interests, has been shown to be a reliable measure of emotional functioning and a good screener of depression. The raw score ranges are from 0 to 25, with higher scores indicating better well-being. The raw score is transformed to 0–100 scale by multiplying by 4. In order to measure possible changes in well-being, a 10% difference can be regarded as a practically significant change (*Topp et al., 2015*). The Czech version (*WHO, 1998*) was used in the present study. The WHO-5 was selected because it is short and valid for the participants' age group (*McDowell, 2010*), robust (*Bech, 2012*), suitable for a mentally healthy sample, and measures positive affect (*Taylor & Brown, 1988*).

## Intervention

The cognitive group was involved in special cognitive tasks such as reaction speed, the time to respond to a simple stimulus (when no decision was involved), the ability to remember information for a short period of time, the ability to split attention between two or more channels of information, the ability to estimate event duration, the ability to retrieve a word from the semantic lexicon, awareness of the channel (source) of knowledge or information, planning as the ability to mentally anticipate the correct way (think ahead) to execute a task, or the ability to grasp quickly the meaning of a visual stimulus. During the first session participants were tested using 11 cognitive tasks. The results of those tasks were used to individualise training programs based on the specific needs of each participant in the cognitive group. Each session of the training program was further adjusted according to the progress of a given participant so that the perceived intensity of the training program is equalized. The training program was validated in *Gigler et al. (2013)*; *Peretz et al. (2011)*; *Shatil (2013)*. The training program was embedded in digital television and allowed participants to interact with the training program in a familiar manner. Television as a low-cost training device was assessed with favourable results in *Boquete et al. (2011)*. Neither touch screens nor sensory controls were used. Participants followed the training program fairly strictly.

The active control group was engaged in leisure activities as if passing time at home. The control group activities were not restricted to non-cognitive tasks and might be cognitively challenging but were strictly different from the cognitive group exercises in that: (a) they were not individualised; (b) they did not follow any specific cognitive training protocol. The control group activities included building a family tree (they had the opportunity to bring their own photos, digitalise them and build their own family tree), image processing and painting, simple gaming using a pentomino application and also Nintendo sport games. Television guided physical exercise was also included in the schedule. During some sessions, participants could select whatever kind of activity appealed to them.

Sessions in both groups were assisted by trained personnel to provide for the proper execution of the intervention. The task of the assistants was to introduce and conclude

the session. The assistants also provided technical support if necessary. All assistants were instructed to interfere in the intervention activities only in situations when participants were not able to continue on their own. The digital television-based training application retained the progress scores of the participants in the cognitive group. All participants kept diaries and wrote comments about how much they liked each intervention session and also about their mood between sessions. They also documented mental activities outside the training protocol that might influence their mood. Usually, at the end of each session, assistants questioned all participants in both groups about their mood status and the mood status was recorded in diaries.

If participants were not able to attend a planned session they could complete it on the same day at a different time. In some special cases, an extra session was added at the end of the intervention program. In the control group, there were no strict programs to follow. In the cognitive group, participants were instructed to adhere to the individualised training program and there was an insistence on adherence.

## Statistical analysis

The IBM SPSS and R statistical packages were used to conduct the statistical analyses. Descriptive statistics were computed to check if there were any significant differences (at the alpha level 0.05) in demographic characteristics between the cognitive group and the control group. The continuous baseline demographic characteristics were analysed using the independent samples $t$-test. The nominal and the ordinal demographic characteristics were tested using the Chi Square statistic.

A mixed effect model for repeated measures was used to evaluate differences in the WHO-5 scores within and between the groups. The dependent variable was the WHO-5 score; the independent variables were Time (pre-test and post-test) as within subject factor and Group (Control or Cognitive), Gender (Male or Female), Education level (High school or University) as between subject factors. The model allowed for the assessment of interventions in post-test scores between the two groups, differences between baseline and post-test scores within each group, and whether there was a significant interaction between the independent variables (whether the impact of one variable was influenced by the level of a second variable). The model also reported the main effect of one independent variable while controlling for the effect of other independent variables. The outcomes were further analysed using marginal means and pairwise comparisons with Least Significance Difference (LSD) test. The mean differences of the within subject variable (Time) were also studied. Differences with $p$-values $<0.05$ were considered statistically significant. In order to measure real changes in WHO-5 score, a 10% difference (out of 100) can be regarded as a practically significant change (*WHO, 1998*).

The effect size (also known as strength of association) was measured by partial eta squared statistics indicating the proportion of variance explained by the independent variable. The guidelines proposed by *Cohen et al. (2002)* for interpreting this value are: 0.01 = small effect, 0.06 = medium effect, 0.14 = large effect.

**Table 1  Baseline characteristics of participants.** None of the statistics were significant at the $p < 0.05$ level.

| | Cognitive Training Group ($N = 37$) | Control Group ($N = 44$) | $p$ value |
|---|---|---|---|
| Age in years (mean ± SD) [range] | 67.8 ± 5.48 [60–79] | 68.0 ± 5.74 [61–84] | 0.880 |
| Female (%) | 62.2 | 59.1 | 0.778 |
| University level education (%) | 48.6 | 34.1 | 0.184 |
| Formal education in years (mean ± SD) [range] | 15.9 ± 2.87 [11–23] | 14.8 ± 3.14 [11–23] | 0.108 |
| MMSE (mean ± SD) [range] | 29.6 ± 0.64 [28–30] | 29.4 ± 0.73 [28–30] | 0.2153 |
| WHO-5 score (mean ± SD) [range] | 66.7 ± 18.50 [24–100] | 65.7 ± 18.37 [8–96] | 0.8132 |

## RESULTS

### Participants in the final analysis

The final statistical analysis included participants who passed the eligibility criteria and finished the whole intervention (i.e., completed all the sessions). The interventions were completed by 117 participants (60 in the cognitive group, 57 in the control group). Due to personal reasons (especially frequent scheduling of sessions and lack of interest) 9 participants (7%) withdrew from the study (3 from the cognitive group (5%) and 6 from the control group (9%). The study eligibility criteria were met by 81 participants (37 in the cognitive group, 44 in the control group) aged 67.9 ± 5.59 [60–84]. None of the participants experienced major technical difficulties with using the television as a training device. Table 1 shows that there were no significant differences in baseline personal characteristics in the two groups.

### Effects of individualised cognitive training and leisure activities on well-being

There was significant interaction between Group and Time, Wilks' Lambda = 0.992, $F(1,73) = 0.563$, $p = 0.018$, partial eta squared = 0.074 (small effect). The interaction indicated that there were significant differences in how participants responded in terms of WHO-5 scores to interventions across the groups regardless of gender and education level. The estimated marginal means revealed that participants in the cognitive group reported lower WHO-5 scores (Pre-test $M = 67.21 ± 3.19$, 95 % CI [60.86–73.56], Post-test $M = 64.13 ± 3.03$, 95% CI [58.09–70.17]) while participants in the control group reported higher WHO-5 scores (Pre-test $M = 66.8 ± 3.03$, 95 % CI [60.77–72.83], Post-test $M = 73.48 ± 2.88$, 95% CI [67.74–79.22]) (Fig. 1). Pairwise comparison of the means using the LSD procedure revealed that the increased WHO-5 scores (Pre-test to Post-test) in the control group were statistically significant (minimum mean difference $M = 6.68 ± 2.78$, 95% CI [1.14–12.22], $p = 0.019$) The reduced WHO-5 scores in the cognitive group were not statistically significant. After the intervention, the control group participants declared significantly higher WHO-5 scores (Post-test) than participants in the cognitive group (minimum mean difference $M = 8.55 ± 3.97$, 95% CI [0.64–16.46], $p = 0.035$).

The mean differences of WHO-5 scores over 8 weeks (Post-test–Pre-test) $M = 6.68 ± 2.78$, 95% CI [1.14–12.22] and M = $-3.08 ± 2.92$, 95% CI [−8.9–2.75] in
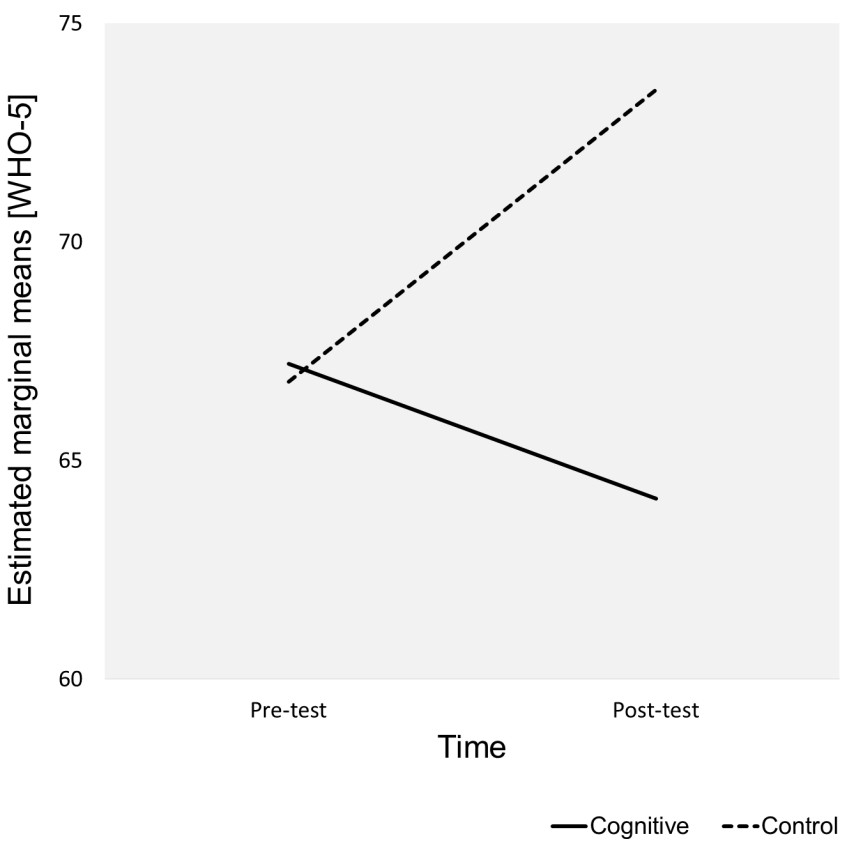

**Figure 1** **Estimated marginal means for Group*Time.** The plot shows the estimated marginal means (WHO-5) for Time*Group.

control group and cognitive group participants, respectively, were below 10% threshold of practical significance (Fig. 2). Therefore, there was no practical difference in the WHO-5 scores in any of the groups.

## Moderating effect of education

There was statistically significant interaction between Time and Group among the participants with only high school education, Wilks' Lambda = 0.876, $F(1,46) = 6.538$, $p = 0.014$, partial eta squared = 0.124 (medium effect). There was no significant interaction between Time and Group among the university graduates. This indicated that high school education participants were the most affected by the different regimen in the control and the cognitive group ($M = 74.53 \pm 3.47$, 95% CI [67.61–81.44] vs $M = 59.93 \pm 4.13$, 95% CI [51.71–68.16]). The marginal means plot in Fig. 3 illustrates the interaction between time and group among university graduates (Fig. 3A) and high school graduates (Fig. 3B). The plot shows that the university graduates declared increased WHO-5 scores in both groups but in the control group the increase was more pronounced. There was an inverse trend among high school graduates with an increase in WHO-5 scores in the control group but a decrease in the cognitive group. The increase in WHO-5 scores among high school
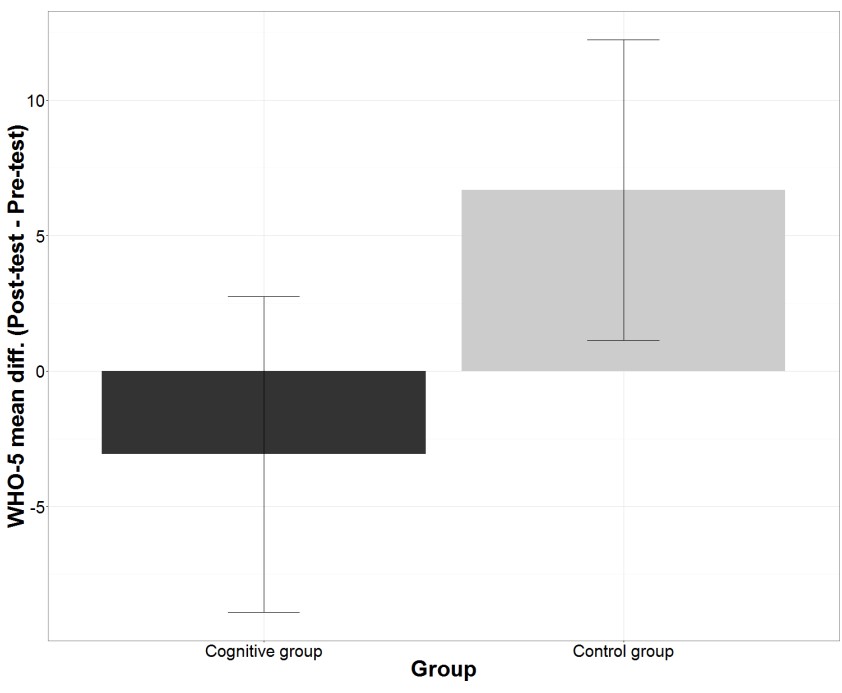

**Figure 2 Mean differences in WHO-5 scores according Group.** The figure shows within subject mean differences in WHO-5 scores with 95% confidence intervals split according the group.

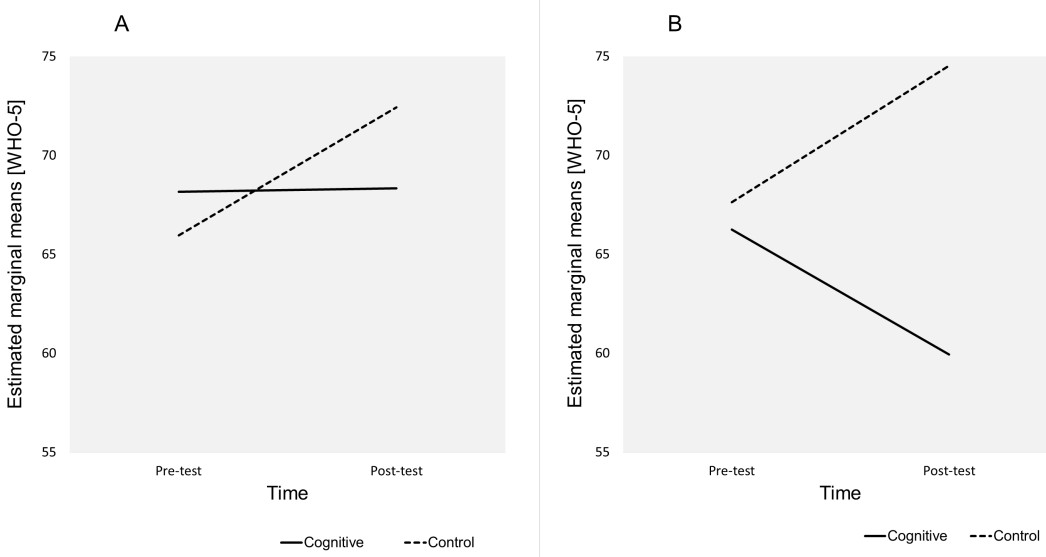

**Figure 3 Estimated marginal means for Time*Education level*Group.** The plots show the estimated marginal means (WHO-5) for Time*Education level*Group. (A) Education level = University. (B) Education level = High school.

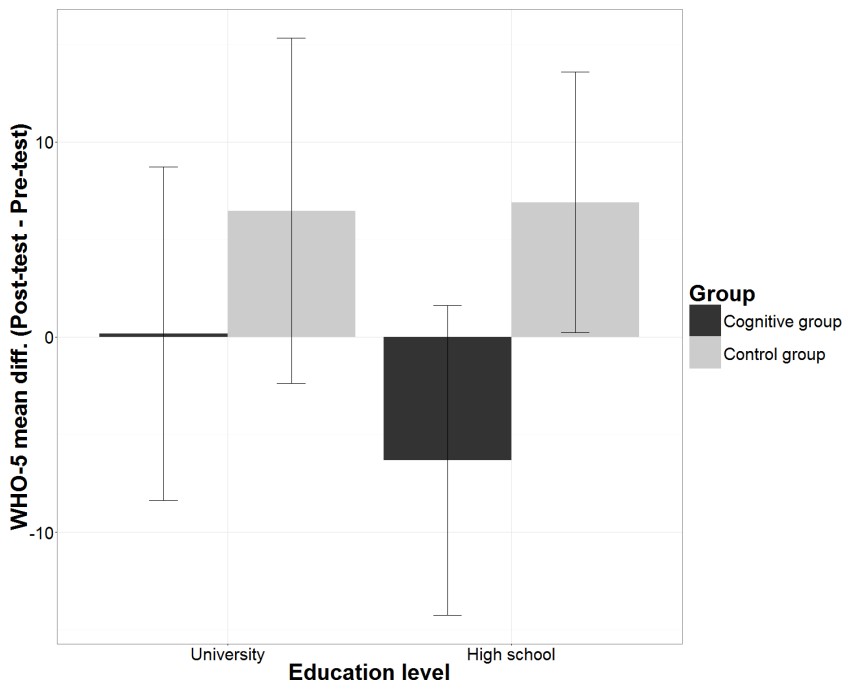

**Figure 4** **Mean differences in WHO-5 scores according Education level and Group.** The figure shows within subject mean differences in WHO-5 scores with 95% confidence intervals and between-subject differences split according the group and the education level.

graduates in the control group is statistically significant, Wilks' Lambda = 0.945, F(1,73) = 4.241, $p = 0.043$, partial eta squared = 0.055 (small effect).

The mean differences of WHO-5 scores over 8 weeks (Post-test–Pre-test) in participants with high school education in the control group were $M = 6.9 \pm 3.35$, 95% CI [0.22–13.57] and WHO-5 scores observed in high school participants in the cognitive group were $M = -6.32 \pm 3.98$, 95% CI [−14.25–1.62]. The mean differences in WHO-5 scores among university graduates were in the control group $M = 6.46 \pm 4.44$, 95% CI [−2.38–15.3] and in the cognitive group $M = 0.17 \pm 4.28$, 95% CI [−8.37–8.71] (Fig. 4). Pairwise comparison of the means using the LSD procedure revealed that after the intervention (Post-test) high school participants in the control group declared significantly higher WHO-5 scores than high school participants in the cognitive group (minimum mean difference $M = 14.594 \pm 5.392$, 95% CI [3.85–25.34], $p = 0.008$). The contrast in mean differences of 14.594 reached the 10% threshold of practical significance in WHO-5 scores. The mean differences among university graduates were not significantly different.

## Moderating effect of gender

There was a statistically significant interaction between Time and Group among female participants, Wilks' Lambda = 0.916, F(1,45) = 4.150, $p = 0.048$, partial eta squared = 0.084 (medium effect). This indicated that women in the control group and the cognitive group responded differently to activities ($M = 73.95 \pm 3.93$, 95% CI [66.13–81.78] vs $M = 64.18 \pm 3.71$, 95% CI [56.79–71.57]). The interaction between Time and Group among male participants was not significant. The marginal means plot in Fig. 5 showed

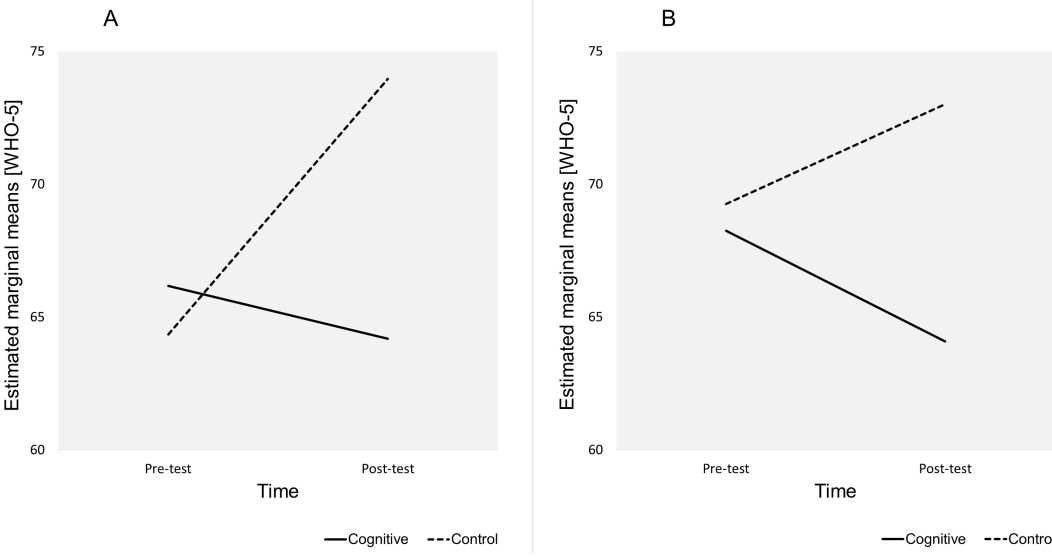

**Figure 5  Estimated marginal means for Time\*Gender\*Group.** The plots show the estimated marginal means (WHO-5) for Time\*Gender\*Group. (A) Gender = Female. (B) Gender = Male.

that women in the cognitive group declared higher baseline WHO-5 scores but lowered their well-being over the 8 week intervention while women in the control group declared an increase in WHO-5 scores (Fig. 5A). A similar pattern could be observed among the male participants apart from the reversed baseline scores (Fig. 5B).

The mean differences in WHO-5 scores in the course of intervention (Post-test–Pre-test) among female participants in the control group were $M = 9.61 \pm 3.79$, 95% CI [2.06–17.16] and in the cognitive group were $M = -1.98 \pm 3.58$, 95% CI [−9.11–5.14]. The mean differences in WHO-5 scores among male participants in the control group were $M = 3.75 \pm 4.07$, 95% CI [−4.35–11.85] and in the cognitive group were $M = -4.17 \pm 4.63$, 95% CI [−13.39–5.06] (Fig. 6). The increase in WHO-5 scores among female participants in the control group was statistically significant, Wilks' Lambda = 0.919, $F(1,73) = 6.431$, $p = 0.013$, partial eta squared = 0.081 (medium effect). The pairwise comparison of the means using LSD procedure revealed that resulting WHO-5 scores at the end of the intervention (Post-test) was not statistically higher compared to females in the cognitive group, $M = 9.77 \pm 5.40$, 95% CI [−0.99–20.50], $p = 0.059$, but the minimum mean difference of 9.77 was nearly at the 10% practical significance threshold.

## DISCUSSION

This research examined the link between individualised cognitive training and the self-rated perception of well-being. The effect of individualised cognitive training was compared with leisure time activities and the moderating effects of gender and education were considered. The first major finding concerns the differences between the interventions when the effects of gender and education were statistically controlled for. The results indicated that individualised cognitive training was not directly associated with improvements

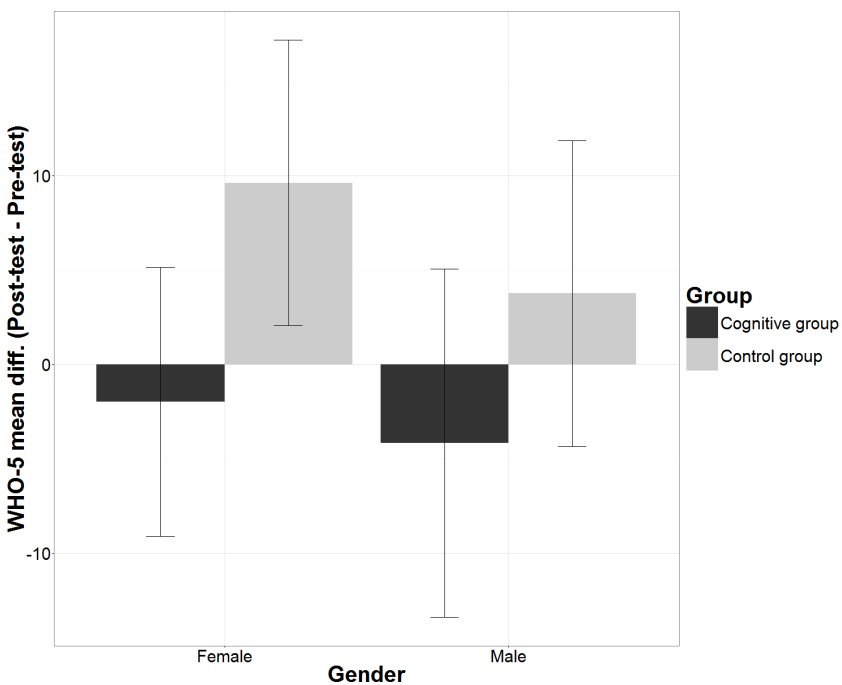

**Figure 6  Mean differences in WHO-5 scores according Gender and Group.** The figure shows within subject mean differences in WHO-5 scores with 95% confidence intervals and between-subject differences split according Group and Gender.

in well-being. Participants in the cognitive group rated their well-being lower after the training intervention, but the decrease in WHO-5 scores was not statistically significant. This finding is in contradiction with the assumed positive effect of cognitive training on well-being and might be contrasted with the effect on cognitive functions where the significant improvements in working memory and executive functions were observed in the cognitive group as previously published in a study based on the same dataset (*Shatil et al., 2014*). The control group, who were only passing time with leisure activities as distinct from strictly individualised cognitive training, showed a statistically significant improvement in self-rated well-being. Although, the improvement was practically not significant, falling below 10% change, it pointed out that involvement in leisure time activities which participants were partly free to choose represents more favourable stimulation to a self-perceived sense of well-being than individualised cognitive training despite improvements in cognitive functions. Apparently, in the perspective of emotional experience, the training routine and the associated stress to perform mentally challenging tasks might reduce the sense of well-being.

The improvement in well-being might be also affected by social factors; participating in a study could be a diversion from daily routine and gave a special meaning to some participants' lives. This corresponds with the concept of the sense of coherence (*Antonovsky, 1987*; *Saevareid et al., 2007*), which suggests that people involved in meaningful activities with social appraisal feel better about life. The social factors, however, could not explain

the differences in self-perceived sense of well-being found in this study as participants in both groups received the same information and the same social support.

The second major finding related to the moderating role of gender and education in the effect of cognitive training and leisure activities on well-being. The results indicated that both gender and education moderated the effect of cognitive training on well-being as compared to the effect of leisure time activities. Females, regardless of their education, reacted to cognitive training and leisure time activities differently. Women engaged in leisure time activities declared significant increase in well-being and after the treatment their level of well-being were higher nearly at the practical significance threshold compared to women in cognitive training. The reaction of males follows the same pattern but did not demonstrate such significance. So, females respond with more life satisfaction to diversion as reported by *Senik (2015)* but when a specific performance is expected their reaction is reversed.

In terms of education, high school graduates, regardless of gender, declare higher well-being then high school graduates after cognitive training in response to leisure time activities. The difference in well-being among high school participants resulted in practically significant changes. University graduates did not respond to such an extent. The decreased perception of well-being among high school graduates in the cognitive group could be explained by lower educated participants feeling less in control as pointed out in other studies (*Ross & Mirowsky, 1989*; *Schulz et al., 1995*) and being uncomfortable under pressure inflicted by an individualised training program. This fact also suggests that education can constitute an effective buffer against the subjective perception of reduced well-being in some situations. These findings were consistent with *Zhang et al. (2011)* who found that that better educated people tend to think logically, rationally, and consistently, see many sides of an issue and thus are not negatively impacted to the same extent by various life events as are less educated people. Similarly, education implies higher income, therefore, access to better healthcare, reputable social status and correspondingly positive social interactions which might also play a role in life fulfilment and satisfaction (*Yakovlev & Leguizamon, 2012*).

Since gender and education proved to moderate the effect of cognitive training, it might be interesting to study the combined effect of gender and education. In this study, however, the number of participants was not high enough to achieve appropriate statistical significance. The analysis of the combined effect of gender and education is therefore referred to further research.

The results may have broad implications for the design of technology based training programmes for elderly people. The aging population in many of the European countries directs attention to cognitive training and a considerable effort has been devoted to developing technology based health strategies to exploit new technological advancements (*European Commission, 2012*; *Scott & Mars, 2013*). Cognitive training delivered via the latest technology has proved to be vital in reversing the adverse effect of cognitive decline (*Kueider et al., 2012*; *Maseda et al., 2013*; *Rizkalla, 2015*; *Shatil et al., 2014*; *Van Muijden, Band & Hommel, 2012*). However, as pointed out in this study, the improved sense of self-perceived well-being is rather associated with involvement in leisure activities than

with improved cognitive function. Especially, being a woman or having only high school education, represents a risk in which cognitive training inflicts a decreased self-perceived sense of well-being instead of maintaining or elevating it. Not addressing the emotional dimension might reduce patients' engagement in the intervention and could be a reason behind the low adherence to training regimens. Low emotional engagement limits the benefits of technology assisted interventions as pointed in (*Barello et al., 2015*; *Graffigna, Barello & Riva, 2013*). Hence, it is difficult then to establish cognitive training as part of everyday life in an elderly population. Reflecting on the emotional experience during cognitive training might help to improve adherence (*Barello et al., 2015*; *Graffigna, Barello & Riva, 2013*). These results have to be verified in a follow up study in order to confirm the effect of cognitive training on the self-perceived sense of well-being over an extended time-frame.

The present research has limitations. The findings were limited due to the fact that the control group participants were involved in leisure time activities. The reason was to avoid a placebo like effect among participants in the cognitive group and motivation distortion among participants in the control group. Therefore, although the programmes in the cognitive and control group were strictly different, from the point of view of the participants the treatment seemed to be to same. Also, the recruitment procedure via advertisements in local newspapers did not give an equal opportunity to all potential participants to take part in the study and the study is limited to participants that responded to the advertisement, thus, the recruitment procedure is prone to selection bias. Participants were also financially motivated to complete the intervention which might have distorted their attitude to the administered activities. The financial reward was €100 approx. (less than €5 per training session) and should have covered travel costs and other out-of-pocket expenses. The results might also be influenced by activities outside the protocol. These activities were recorded in participants' diaries but were not included in the statistical analysis. Correspondingly, this study did not examine the influence of other potentially important factors that might account for the statistically significant changes. The environment in which cognitive training was conducted or the examination of specific age groups are some examples. Moreover, the study also focused on healthy individuals capable of living on their own. The results might differ if elderly people with mental disorders or depression were considered.

## CONCLUSIONS

In conclusion, although there are several studies proving the effect of cognitive training on cognitive function, little attention has been paid to the effect of cognitive training on self-perceived well-being. This study administered individualised cognitive training and investigated the effect on emotional experience as measured by the WHO-5 index. In comparing the outcomes for the cognitive training group and the active control group the study revealed that the self-perceived sense of well-being is more likely stimulated by leisure activities. The results showed that cognitive training was not directly associated with improvements in well-being. Gender and educational attainment appeared to play a role in the effect of cognitive training on well-being as females and participants with

only high school education declared a decreased self-perceived sense of well-being after the intervention. The study opens new space for research in which cognitive function and well-being are treated as different phenomena. A different approach to stimulating cognitive function and a sense of well-being might help to improve adherence to the cognitive training regimen. There were several limitations that should be addressed in the follow up study and in future research.

## ACKNOWLEDGEMENTS

The authors would like to thank all Vital Mind project members from the Philips Consumer Products Innovative Lab, Czech Technical University in Prague, University of Genoa, Czech TV, University of Dundee, University of Hradec Králové, Goldsmith College in London, and CogniFit, the coordinator, for their valuable contribution to the project. The long term development plans of UHK and FNHK are acknowledged.

### Funding

This study was partially funded by the Czech Science Foundation project DEPIES no. GA15-11724S and the project Vital Mind, European Union 7th Framework Programme, project number ICT-215387. The funders had no role in study design, data collection and analysis, decision to publish, or preparation of the manuscript.

### Grant Disclosures

The following grant information was disclosed by the authors:
Czech Science Foundation: GA15-11724S, ICT-215387.

### Competing Interests

The authors declare there are no competing interests.

### Author Contributions

- Vladimír Bureš conceived and designed the experiments, performed the experiments, wrote the paper, reviewed drafts of the paper.
- Pavel Čech conceived and designed the experiments, performed the experiments, analyzed the data, wrote the paper, prepared figures and/or tables, reviewed drafts of the paper.
- Jaroslava Mikulecká conceived and designed the experiments, analyzed the data, contributed reagents/materials/analysis tools, wrote the paper, prepared figures and/or tables, reviewed drafts of the paper.
- Daniela Ponce performed the experiments, reviewed drafts of the paper.
- Kamil Kuca analyzed the data, contributed reagents/materials/analysis tools, reviewed drafts of the paper.

## Human Ethics

The following information was supplied relating to ethical approvals (i.e., approving body and any reference numbers):

The European Commission granted ethical approval to carry out the study within the scope of the Vital Mind project by accepting the Ethical Road Map report (deliverable D8.6.1). The institutional ethical approval was granted by the chancellor's office of the University of Hradec Kralove (Vital Mind ref. no. 215387). Note to the editor: the ethical approval in the University of Hradec Kralove is granted through the Chancellor's office as a part of general approval that considers all legal aspects of the study (including the ethical code of conduct). The general approval is referenced by the project number. Since we do not have the ethical approval number, we provide chancellor's declarations that the study was granted ethical approval during the Vital Mind project duration. Also, the Ethical Roadmap Report approved by the European Commission is attached as Data S1.

## Data Availability

UHK Explorer

Open Access

http://explorer.uhk.cz/pc/www/peerj/data.zip.

## Supplemental Information

Supplemental information for this article can be found online at http://dx.doi.org/10.7717/peerj.2785#supplemental-information.

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
