# Peer review of "The effect of cognitive training on the subjective perception of well-being in older adults"

_PeerJ, doi:10.7717/peerj.2785_

## Round 0.1 · original submission · Major Revisions

Dear Pavel,

Thank you for your submission to PeerJ. We have 2 substantial reports from reviewers 2 and 3 (see below). Please respond them in your rebuttal and resubmission

Reviewer 1 ·

Basic reporting

The article was write in good and clear English, and with a sufficient introdution and with a good discussion of the results.

Experimental design

The study was made in the good of practicle principles and with consistent statistical tests and with results consistent.

Validity of the findings

The results are robust and well controlled.

Additional comments

A article consistent and important to the clinical practice.

Reviewer 2 ·

Basic reporting

The manuscript is an original research about the effect of a cognitive training programme on the well-being perceived by an elderly sample. By using a random design, the results haven’t shown that the intervention improve the subjective well-being. The manuscript is well written and accordingly to the PeerJ standards.

Experimental design

The research is well designed and the question is relevant and meaningful, given the target population. The Methods are described with sufficient information to be reproducible and the results are properly presented.

However there are some concerns that will need to be addressed by the authors:
1. The Introduction is not sufficient to demonstrate how this research fits in the field of knowledge about what is already known.
2. During the manuscript “well-being” also appears as “happiness” and “quality of life” indiscriminately. These terms should be clearly defined as well as distinguished in the Introduction.
3. A clear definition of the research question is also missing.
4. Authors use "gender" and "education" as variables in the association between the intervention and the outcome. However this choice needs to be better justified in the Introduction, in order to specifically address the topic of the research.
5. The methods used to recruit the sample (via advertisement) should appear as a limitation of the study since they are prone to important selection bias.
6. In the Discussion, authors should have emphasized the novelty as well as the significance of the results.
7. This latter section also needs to be reformulated in order to address the specific research question (the effect of cognitive training on the well-being) and to interpret the results in light of the literature in the same field. Some of the statements displayed in the Discussion are not derived nor supported by the presented results (see e.g., lines 307-308).
8. In line 333, authors should discuss this statement in relation with the literature/other relevant studies.
9. In the Discussion, line 375, other variables such as “blood pressure” and “stress” are referred. The relation of these variables with the well-being needs better clarification.
10. In the Conclusion, it is referred that the “well-being appeared to be influenced by other variables other than mere improvements in cognitive function”. However since this improvements were not measured in the study, the authors can not conclude about their influence.

Validity of the findings

A 0.1 alpha level was used which refers to a 10% error probability, but this choice is assumed by the authors as weakness of manuscript.

There are other important questions that deserves to be addressed:
1. The sample size is not clear (140, 126 or 117 elderly?).
2. In Table 1, the range of education years varies from 11 to 46 years. This is neither reasonable nor informative. Formal and non-formal education should be analysed separately within and between the groups.
3. Authors refer the use of MMSE cut-off of 23. However it is not clear if this cut-off is derived from any Czech validation.
4. It is referred that the “participants were financially motivated to complete all training and assessment sessions”. To add transparency to the findings, the kind of financial motivation should be clarified. Was it only in the cognitive group? Besides, it should also be stated as a limitation of the study and of findings validity.
5. It is referred (line 372) that a higher education is “a buffer” in the studied association. However, the results show that the group with more university graduates (56.7% vs 32.2% in the control group) was the one that exhibited a decrease in well-being. How could you explain it?

Additional comments

Other aspects should also be reviewed:
1. In line 67, “etc” should be replaced, and references that support the stated information would be indicated.
2. The lines 62-66 should be placed in Materials & Methods section.
3. The lines 84-85 do not have association with the previous and need revision.
4. Table 1 is Results material.
5. Revise line 249 in order to make it congruent with the education levels presented in Table 3.

Reviewer 3 ·

Basic reporting

• Introduction is poorly structured and confusing. A detailed description of the current evidence regarding different methods of cognitive training should be provided first. Only after justifying the need for these kind of studies should the authors refer the objective of the study. Details regarding participants and intervention should be saved for the methods section. More supporting references should be included after each statement (e.g.: what is the evidence that supports the following sentence, particularly regarding the impact of aging on perception: “Even normally aging older adults experience significant impairment in attentional tasks, long-term and working memory tasks, perception, speed of processing, etc.”; or which studies are referenced when the authors say “Concerning education, recent studies show that…”).
• Which version of the MMSE was used? If a translated version was used, it should be referenced. Culturally adaptation could have also influenced the cut-off points. Was education level considered for the cut-off? Usually, it does. How was the criterion MMSE ≥23 established? Did the participants have to touch the television during the training? If so, subjects unable to touch the television were also excluded.
• L123 – Why do you refer “focus” group? This usually means a group were subjects discuss a topic.
• Shouldn’t participants in both groups be matched according to formal and non-formal education as well? It influences the participants’ cognitive reserve and potential for cognitive gains. In L131-132 it is not clear that they were also matched for education.
• I believe that the participants’ characteristics should be in the result section.
• Which is the bibliographical reference for the Czech version of WHO-5
• Abstract should contain some of the values that resulted from the statistical tests. Language should be clearer and not ambiguous. For example, it should be stated that participants where cognitively healthy. Results should also be clearer and should not repeat information.
• Please check paper for orthographic errors (e.g. Line 446: “Experimantal”)

Experimental design

• How did the participants in the intervention group interact with the TV to transition between tasks? Were the selected tasks chosen considering the tasks they had more difficulties or according to their interests and motivations? If the tasks that were chosen were the tasks in which the participants failed more times, this could negativity impact mood and well-being! Constantly failing is not a pleasant experience. Was there any progression in difficulty during the protocol?
• I consider that the “control” group might have had a more positive experience with the intervention than the “intervention” group, possibly influencing the outcomes. Participants in the intervention group had to perform very specific cognitive tasks over and over again, while the control group had to perform fun and diverse leisure activities. Activities which I believe also are cognitive stimulating (e.g. executive functions are needed for the pentomino). Furthermore, it included physical activities which can also improve well-being and even cognitive function. Also, probably (it is not well explained), the assistants had a more active role in the control group than in the intervention group, engaging with participants and possibly making their experience more supported and enjoyable. Therefore I am not surprised that the control group had better results than the intervention group regarding subjective well-being. This should be well discussed in the discussion section and description of the intervention/control conditions should be more detailed (methods section).
• How usual was the assistants question regarding mood after sessions? Was this approach similar for all participants? Don’t you believe that supervising what participants write in their diaries influences what they write and how they feel?
• Explain L188-190: “In the control group there were no strict programs to follow. In the cognitive group, participants were instructed to adhere to the individualised training program and adherence was supported by the trained assistants”. The participants of the control group had no financial motivation? How could these differences influence the outcomes?
• Was any follow-up considered?

Validity of the findings

• It is not clear why education was assessed firstly in total years of formal and non-formal education and secondly, in the mixed model, in university or high school.
• It is not clear if gender was inserted as a covariate in all tests. Why was it used considering that there were no differences between groups? Why only gender and not e.g. MMSE?
• If 140 subjects were recruited and 117 completed the intervention protocol, why does the abstract state that the study comprised 126 participants?
• Results are not clear. In the abstract it is stated that “There was no statistically significant change in the WHO-5 index for the cognitive group. In the control group, there was a statistically significant increase in subjective well-being.” But in the results you say that “There was a statistically significant main effect for time…with both groups showing an increase in WHO-5 scores”. Results are very confusing. The values included in the tables do not help much. Perhaps plots for each model should be provided.
• Additional participant characteristics should be provided in the results section, particularly regarding education (university vs high school), lifestyle and clinical condition.
• Discussion is poorly written, with repetition regarding result reporting. Improvement is required particularly regarding the justification for group differences and the influence of education, as well as discussion of possible bias. For example, was there any documentation of other activities performed by the participants outside the protocol? Could their participation in other significant activities have influenced well-being?

Additional comments

The article is interesting, but some limitations have been identified.

---

## Round 0.2 · Minor Revisions

Dear Pavel,

Thank you for your submission to PeerJ. As you can see, there are substantial comments from Reviewer 2 which you should address.

Reviewer 2 ·

Basic reporting

The manuscript is an original research about the effect of a television-based cognitive training programme on the well-being perceived by an elderly sample. Following the suggestions of the first review, the authors have introduced new and relevant information to the manuscript, as well as major modifications concerning the eligibility of the participants and statistical analysis. This is practically a new manuscript, and there are aspects that still deserve to be addressed and/or clarified.

Experimental design

1. Concerning the Abstract:
1.1. On the Methods, the other measures used should be stated.
1.2. On the Discussion, since the data concerning the effect of the programme on cognitive functions are not shown, authors should not focus these functions.

2. Concerning the Introduction:
2.1. A clear definition of the research question is still missing. The authors have studied the impact of the cognitive programme (intervention) in the self-perceived well-being (outcome - measured by the WHO-5), comparing two groups. Once you have not originally compared groups of different gender and/or educational attainment, the formulation “What will be the difference in the self-perceived sense of well-being of elderly people involved in individualised cognitive training in regard to gender and education?”, should be reviewed.
2.2. The state of the art should not be mixed neither with the research question (lines 57-59), nor with the aim (lines 95-99) or other aspects inherent to your work (e.g., line 111). Place these elements on the end of the Introduction in order to give a more coherent context to the work.
2.3. In lines 98-99, is important to identify these “leisure time activities” as your control group.
2.4. In line 105, treatment is mentioned. The authors could not refer a treatment for the normal changes of ageing. “Prevention” would be more appropriate.
2.5. From line 127 on, authors are gathering evidence for aspects that are not their study issue (the effect of education on cognition). Instead, some evidence supporting the technology of the training programme (digital television) will enrich your work. In fact, part of the information in the Discussion (lines 438-443) is literature review, and should be better placed in the Introduction section.
2.6. The lines 134 and 135 should be supported by literature references.

Validity of the findings

An alpha level of 0.05 was used in the reviewed version of the manuscript, which strengthen the results. However, and concerning the validity of the findings, some aspects should be clarified.

1. The methods used to recruit the sample (via advertisement) should appear as a study limitation not only because of the coverage of the bulletin used, but because those who volunteered are different (e.g., have increased need for approval) from those who did not (volunteer bias). This aspect also limits the external validity of the findings, and should be expressed as a limitation.
2. Data and norms from Crum et al. (1993) are related to USA population, so the cut-offs may not be adapted to your population. It could be an acceptable solution if there are no Czech validations, and this must be reported.
3. In line 213 is referred that the training programs were individualized. Does the customization of the sessions imply different degrees of difficulty that could account for the differences in the perceived well-being (between the groups)?
4. In lines 135-136, the use of diaries is mentioned. Did you analyse their content in order to explain the differences in results?
5. According to the Results section, the eligibility criteria were applied after the end of the study. This is neither reasonable nor correct given your study design. It means that you have selected the participants after the group allocation. Can you clarify this aspect?
6. In lines 443-445, caution should be taken in the generalization of the findings from the present study. Additionally, adding some previous research reporting the same conclusion would be important.

Additional comments

Other minor aspects should also be reviewed:

1. On the Background section of the Abstract, the two first ideas seems disconnected. Maybe the use of “on the other hand” would be more appropriate than “however”.
2. In line 73, please correct “mediation” to “meditation”.
3. Notice that the research of Bier et al. (2015) did studied the effect of cognitive training by using a control group of healthy elderly (in addition to the MCI patients). Taking this into account, the line 94 should begin with “the majority of these studies” (instead of “none of the studies”), in order to give more accuracy to the literature revision.
4. The lines 120-123 are establishing a link that is not coherent. Authors should revise it.
5. The lines 178-179 are Results material.
6. The data on all participants, presented in the abstract, should also be in the Results.
7. In line 413, authors should be cautious when referring “the intervention”, because what is referred is the study comparator, i.e., the sessions of the control group.

Reviewer 3 ·

Basic reporting

Introduction is more complete

Experimental design

Methods have been clarified

Validity of the findings

Findings are valid

Additional comments

The article has been successfully modified.

---

## Round 0.3 · accepted · Accept

Dear Pavel

When in production, please pay attention to the minor suggestions from reviewer 2.

Reviewer 2 ·

Basic reporting

The manuscript is a secondary analysis of data from a randomised controlled trial that addresses the effect of a validated cognitive training programme on the well-being perceived by a sample of healthy community-dwelling older adults. The results have shown that the intervention did not improve the subjective well-being.

Experimental design

The experimental design was clarified and it is now assumed that this is a secondary analysis. I have no further comments regarding this section.

Validity of the findings

I have no comments.

Additional comments

Some minor aspects:
1. On line 33 of the Abstract, add the measure WHO-5 index after the results.
2. On line 68 of the Introduction, replace or exclude the reference from Eiser & Morse (2001), this is related to childhood.
3. On line 91, the sentence “as evidenced by search in PubMed databases dated to October 2016” is unnecessary. Consider its exclusion.
4. Lines 143-148 are about ethics procedures and should be stated at the end of the section in order to make it more clear.
5. Consider the exclusion of the following “published in a widely prestigious international journal” on page 195. Again this seems unnecessary to support the previous idea.
6. The legend of Table 1, should be shortened. Some of the information will be better placed in the text.
7. On line 324, review the meaning of “…the most effected…”.
8. On lines 458-459 references should be added to support this statement.
9. The manuscript should not end with limitations.